# (GIGA)byte

DATA RELEASE

# Near chromosome-level and highly repetitive genome assembly of the snake pipefish *Entelurus aequoreus* (Syngnathiformes: Syngnathidae)

Magnus Wolf[1,2,3,*], Bruno Lopes da Silva Ferrette[1], Raphael T. F. Coimbra[1,2], Menno de Jong[1], Marcel Nebenführ[1,2], David Prochotta[1,2], Yannis Schöneberg[1,2], Konstantin Zapf[1,2], Jessica Rosenbaum[2], Hannah A. Mc Intyre[2], Julia Maier[2], Clara C. S. de Souza[2], Lucas M. Gehlhaar[2], Melina J. Werner[2], Henrik Oechler[2], Marie Wittekind[2], Moritz Sonnewald[4], Maria A. Nilsson[1,5], Axel Janke[1,2,5] and Sven Winter[1,2,6,*]

1 Senckenberg Biodiversity and Climate Research Centre (BiK-F), Frankfurt am Main, Germany
2 Institute for Ecology, Evolution, and Diversity, Goethe University, Frankfurt am Main, Germany
3 Institute for Evolution and Biodiversity, University of Münster, Münster, Germany
4 Senckenberg Research Institute, Department of Marine Zoology, Section Ichthyology, Frankfurt am Main, Germany
5 LOEWE-Centre for Translational Biodiversity Genomics (TBG), Frankfurt am Main, Germany
6 Research Institute of Wildlife Ecology, University of Veterinary Medicine, Vienna, Austria

**Submitted:** 20 October 2023

**\*** Corresponding authors. E-mail: Magnus.Wolf@senckenberg.de; Sven.Winter@senckenberg.de

Preprint submitted at https://doi.org/10.1101/2023.12.12.571260

## ABSTRACT

The snake pipefish, *Entelurus aequoreus* (Linnaeus, 1758), is a northern Atlantic fish inhabiting open seagrass environments that recently expanded its distribution range. Here, we present a highly contiguous, near chromosome-scale genome of *E. aequoreus*. The final assembly spans 1.6 Gbp in 7,391 scaffolds, with a scaffold N50 of 62.3 Mbp and L50 of 12. The 28 largest scaffolds (>21 Mbp) span 89.7% of the assembly length. A BUSCO completeness score of 94.1% and a mapping rate above 98% suggest a high assembly completeness. Repetitive elements cover 74.93% of the genome, one of the highest proportions identified in vertebrates. Our demographic modeling identified a peak in population size during the last interglacial period, suggesting the species might benefit from warmer water conditions. Our updated snake pipefish assembly is essential for future analyses of the morphological and molecular changes unique to the Syngnathidae.

**Subjects** Genetics and Genomics, Evolutionary Biology, Marine Biology

## INTRODUCTION

The snake pipefish *Entelurus aequoreus* (Linnaeus 1758) is a member of the family Syngnathidae, which currently includes over 300 species of seahorses and pipefishes [1]. *E. aequoreus* shares typical features with other pipefishes, such as the unique, elongated body plan and fused jaws [2]. However, unlike most pipefishes, which are found in benthic habitats, the snake pipefish inhabits more open and deeper seagrass environments and occurs even in pelagic waters [2]. They are ambush predators on small crustaceans and

other invertebrates, thereby indirectly contributing to the overall biodiversity and stability of these fragile habitats [3]. Adult snake pipefishes are poor swimmers equipped with small fins. They rely on their elongated, thin bodies for crypsis in eelgrass habitats [4–6].

The snake pipefish historically ranged from the waters of the Azores northwards to the waters of Norway and Iceland and eastward to the Baltic Sea [7]. However, since 2003, the species has expanded its distribution [8] into the Arctic waters of Spitsbergen [9], the Barents Sea, and the Greenland Sea [10]. Simultaneously, population sizes seem to increase within its former range, as indicated by substantially increased catch rates [11, 12]. Several factors have been proposed to cause this expansion and population growth, including rising sea temperatures, an increased potential for long-distance dispersal of juveniles via ocean currents [4, 7], and an increased reproductive success facilitated by the dispersal of invasive seaweeds [6, 8–10, 13]. The latter explanation has been confirmed by local field experiments in the northern Wadden Sea, suggesting a mutual co-occurrence of the invasive Japanese seaweed (*Sargassum muticum*) and the snake pipefish [5]. Studies based on mtDNA marker regions did not discern any population structure thus far and suggest a previous population expansion in the Pleistocene, around 50–100 thousand years ago (kya) [6]. Yet, a comprehensive analysis of demographic events is better conducted using genomic data, thus requiring a high-quality reference genome, ideally of the same species or at least a closely related one.

Previously, genomes of Syngnathidae have been used to study the evolution of highly specialized morphologies and life-history traits unique to pipefishes and seahorses [14–16]. For instance, the transition to male pregnancy was associated with major genomic restructuring events and parallel modifications of the adaptive immune system. There is a remarkable variability in genome sizes within the family, with estimates ranging from 350 Mbp to 1.8 Gbp [14]. The major shifts in body shape are assumed to be related to gene-family loss and expansion events, along with higher rates of protein and nucleotide evolution [16]. Genomic data obtained using a direct sequencing approach of ultra-conserved elements improved the understanding of the phylogeny of pipefishes [15] and identified a likely radiation of the group in the waters of the modern Indo-Pacific Ocean. Nevertheless, high-quality genomes of Syngnathidae are only available for a few species. According to the NCBI genome database, only 7% of the known species diversity has genome sequences available.

A draft genome of the snake pipefish was previously assembled using a combination of paired-end and mate-pair sequencing techniques, yielding an assembly with low continuity (N50 3.5 kbp, BUSCO C: 21%) and a large difference between the estimated and assembled genome sizes (1.8 Gbp vs. 557 Mbp) [14]. To obtain a higher quality, near chromosome-scale genome assembly for the snake pipefish, essential for future population, conservation, and evolutionary genomics studies of fish, we used long-read sequencing technologies. This allowed us to gain insights into the genetic properties of the species and to perform demographic analyses based on the Pairwise Sequentially Markovian Coalescent (PSMC) framework [17]. The data generation and analyses presented here were conducted during a six-week master course in 2021 at the Goethe University, Frankfurt am Main, Germany. The concept of high-quality genome sequencing in a course setting has so far yielded three reference-quality fish genomes and has proven to be a successful approach to introducing the technology to a new generation of scientists [18–21].



**Table 1.** Summary statistics of the snake pipefish reference genome. The table includes information for (A) the raw read sequencing, (B) the scaffold- and contig-level *de novo* assembly, and (C) the BUSCO completeness statistics.

| (A) Raw read statistics | | |
|---|---|---|
| No. short reads | 264,111,731 | |
| Mapped short reads (%) | 99.53 | |
| Mean short read coverage (*x*) | 23 | |
| No. long reads | 130,590,372 | |
| Mapped long reads (%) | 98.61 | |
| Mean long-read coverage (*x*) | 205.2 | |
| **(B) Assembly statistics (scaffold/contig)** | | |
| No. scaffolds/contigs | 7,387 | 7,473 |
| No. scaffolds/contigs (>50 kbp) | 466 | 526 |
| Scaffold/contig L50 | 12 | 14 |
| Scaffold/contig N50 (bp) | 62,341,166 | 45,010,074 |
| Total length (bp) | 1,662,053,046 | 1,662,035,846 |
| GC (%) | 38.87 | 38.87 |
| No. of N's per 100 kb | 1.03 | 0.0 |
| Heterozygosity (%) | 0.387 | |
| Total interspersed repeats (bp) | 1,237,929,559 (74.93%) | |
| **(C) BUSCO completeness** | | |
| Clade: *Actinopterygii* | C: 94.1% [S: 92.6%, D: 1.5%] | |
| | F: 2.0%, M: 3.9% | |
| | n: 3,640 | |

BUSCO: Benchmarking Universal Single-Copy Orthologs (*65*); C, complete; S, single copy; D, duplicated; F, fragmented; M, missing.

## RESULTS AND DISCUSSION

### Genome sequencing and assembly

PacBio's continuous long read (CLR) technology generated 401 Gbp of long-read data in ~60 million reads with an N50 of 7.9 kb (Table 1). Illumina sequencing yielded 38 Gbp of standard short-read data in approximately 257 million reads with a mean length of 148 bp after filtering. Sequencing of the Omni-C library generated 54.7 Gbp of raw short-read data.

The snake pipefish's genome was assembled *de novo* to a total size of 1.7 Gbp. It consisted of 2,204 scaffolds, with a scaffold N50 of 62 Mbp and an L50 of 11 (Table 1, Figure 1A). The finalized assembly has 1.0 Ns per 100 kbp and a GC content of 38.84%. Our BUSCO (RRID:SCR_015008) completeness assessment resulted in 94.1% complete core genes, based on the *actinopterygii_obd10* set, showing the high completeness of the assembly. Both long- and short-read data mapped onto the assembly with high mapping rates of 98.6% and 99.5%, respectively. HI-C mapping resulted in 28 larger scaffolds (Figure 1B), indicating the near-chromosome level of the *de novo* assembly. This result aligns with past karyotype estimations of other pipefish and seahorses, predicting 22 and 22-24 chromosomes, respectively [22–24]. The rest of the genome comprises only smaller scaffolds and contigs, which may result from the high amounts of repetitive regions, as detailed in the following section. Our Blobtools (RRID:SCR_017618) analysis of both long- and short-read data (Figure 1C and D) found no apparent signs of contamination. However, we detected and removed background noise of unknown origin in both datasets.

Variant calling identified ~301 million sites (including monomorphic sites), with ~1.3 million found to be biallelic. Genome-wide heterozygosity was determined to be 0.387%, which is in line with other fish species [25, 26]. The GenomeScope (RRID:SCR_017014) results based on short reads suggested a haploid genome size of 1.15 Gbp and an expected



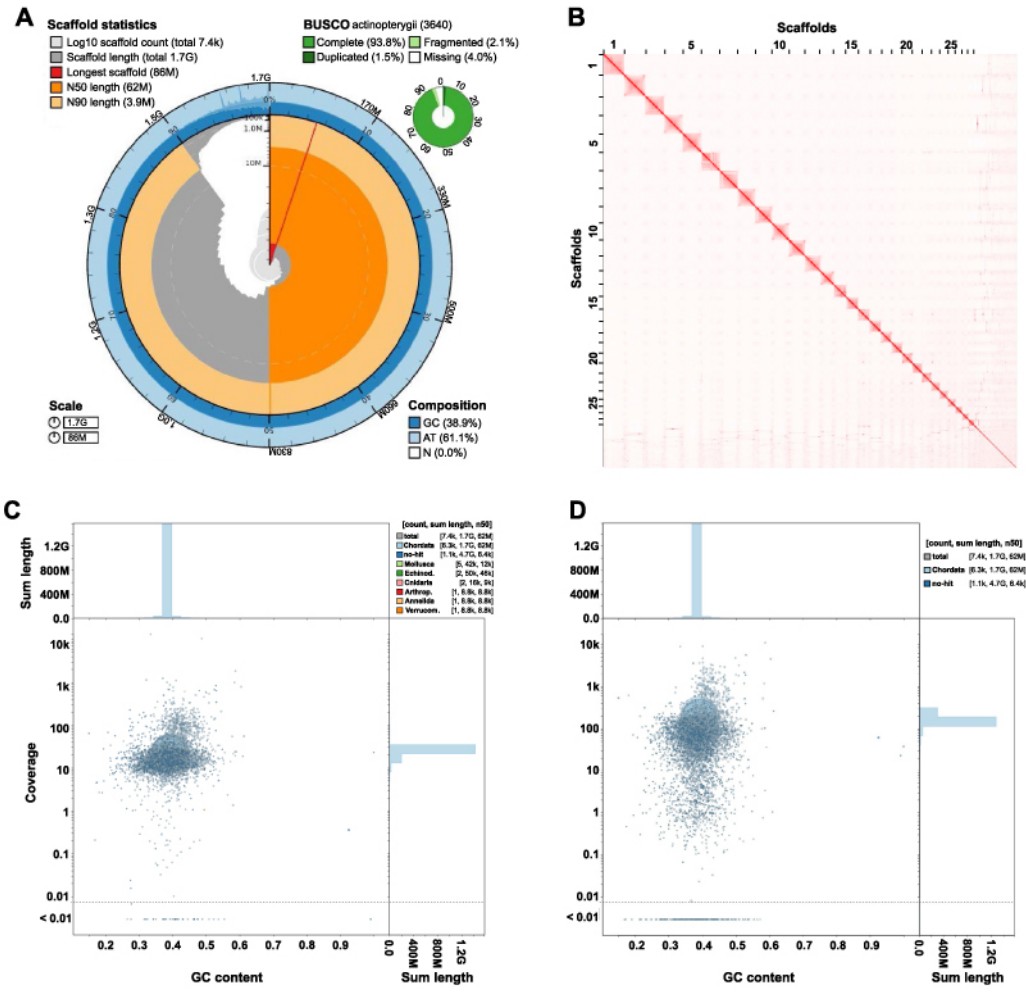

**Figure 1.** Assembly characteristics and quality assessments of the *de novo Entelurus aequoreus* genome. (A) The snail plot summarizes different assembly properties. Scaffold statistics are depicted in the innermost circle, and colors red to orange represent the longest scaffold N50 and N90, respectively. GC composition is shown in the outer blue circle. BUSCO completeness statistics are depicted in the small green circle. (B) Omni-C contact density map indicating 28 larger scaffolds and the near-chromosome level of the assembly. (C,D) The BlobPlot analysis compares GC content (*x*-axis), assembly coverage (*y*-axis), and taxonomic BLAST assignments of contigs (color) for both the Omni-C short reads (C) and PacBio long reads (D).

genome-wide heterozygosity of 1%. These estimates were around 362 Mbp shorter and 0.57% more heterozygous than the final assembly. This, again, might be explained by the high repeat content of the genome.

## Annotation

In total, 0.9 Gbp, or 74.93%, of the entire assembly, were identified as repetitive during our *de novo* repeat-modeling (using RepeatModeler, RRID:SCR_015027) and repeat-masking (using RepeatMasker, RRID:SCR_012954) as shown in Figure 2. This high repeat content contrasts with other fish genomes [27]. However, it is similar, although at a smaller scale, to the closest relative, *Nerophis ophidion* (65.7%) [14], and other syngnathid fish genomes, such as seadragons [28]. The first draft assembly of the snake pipefish had a repeat content of



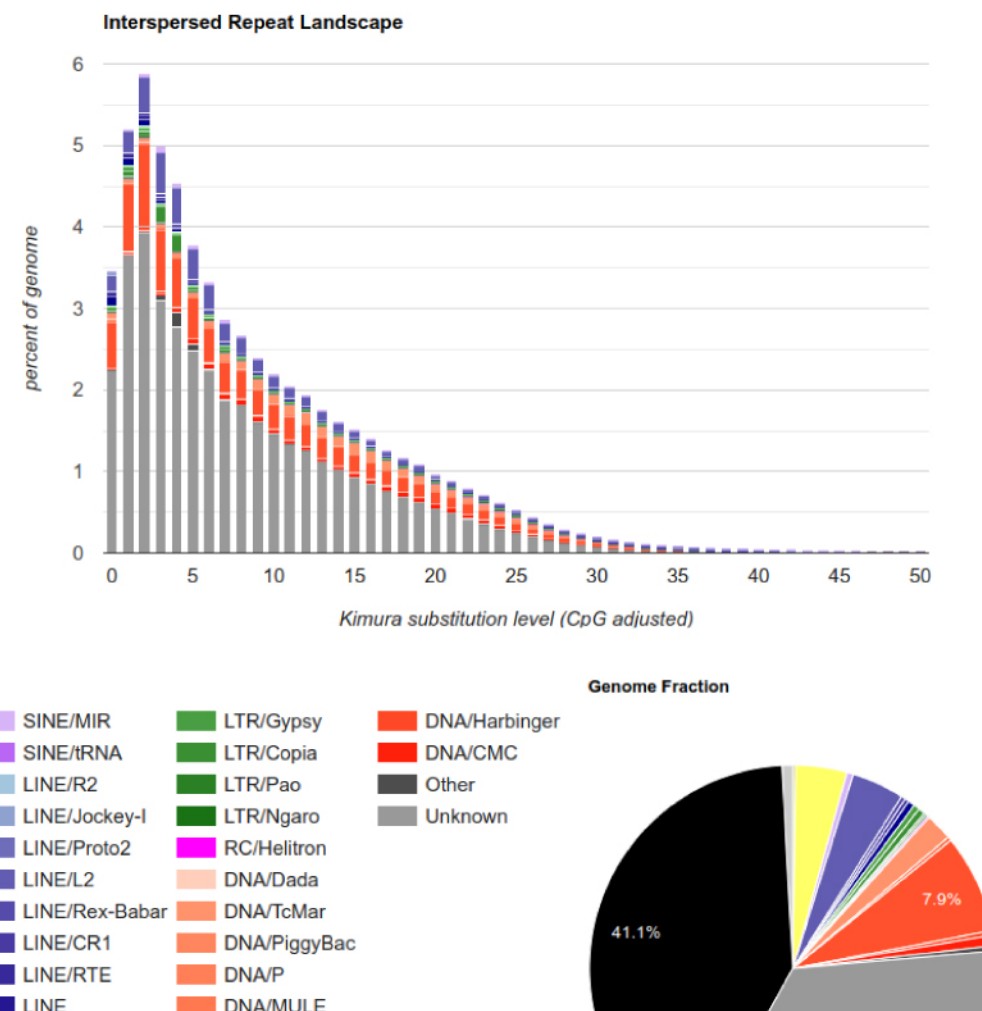

**Figure 2.** Repeat landscape of the *de novo Entelurus aequoreus* genome. Colors represent different types of RE, and gray areas indicate unclassified types of repetitive regions.

57.2% [14], and our improved long-read assembly identified 17.7% additional repeats that were missing from the previous assembly [14]. So far, among vertebrates, only the lungfish *Neoceratodus forsteri* [29] has more transposable elements (TEs) than the snake pipefish.

The annotation of the genome, featuring *de novo* and homology-based identification approaches, resulted in 33,202 genes with an average length of 13,828 bp. Each gene had, on average, 7.32 exons and 6.25 introns with average lengths of 188 bp and 2,240 bp, respectively. In total, we identified 243,038 exons and 207,467 introns within our annotation. The total number of genes is ~30% higher compared to other annotated genomes in the order Syngnathiformes, such as 23,458 for the tiger tail seahorse (*Hippocampus comes*) [16]



or 24,927 for the greater pipefish (*Syngnathus acus*) [30] made by the NCBI Eukaryotic Genome Annotation pipeline. Notably, as these two genomes are considerably smaller (492 Mbp and 324 Mbp, respectively), we can assume that the large-scale genome increase in this species included many coding sequences. The high content of repetitive regions and the lack of transcriptomic data might also have increased the number of false positive gene-calls; however, our BUSCO completeness analysis of the predicted proteins resulted in 82.6% complete sequences, with only 6.8% duplicated ones. Additionally, 5.3% of the coding sequences appeared fragmented, and 12.1% were missing from the *actinopterygii_obd10* OrthoDB set. Our functional annotation resulted in hits for 89% of the predicted proteins.

## Demographic inference

The demographic inference analysis of the snake pipefish genome (Figure 3) using the PSMC framework [17] traced population changes over the past 1 million years. Given the chosen substitution rate and generation time, there was a steady increase in the effective population size ($N_e$), starting at 15 thousand individuals 1 million years ago, which peaked at an $N_e$ of 250 thousand individuals 100 kya. Thereafter, $N_e$ decreased until reaching 30 thousand individuals at 10 kya and stagnated until the end of the model. The previously suggested population expansion during the Pleistocene (50–100 kya) was therefore confirmed by this model. However, the population expansion was followed by another population decline that was not resolved by Braga Goncalves *et al.* [6]. Our result may point to a conclusion different from that drawn by the authors. This is because the snake pipefish might have inhabited a comparably small population during the Holocene and only recently expanded its distribution. This expansion resulted in a large population with a high degree of homogenization, as observed by Braga Goncalves and colleagues [6]. Given that the presented peak in population size parallels with the last interglacial period between the Penultimate Glacial Period (135–192 kya [31]) and the last glacial period (present – 20 kya [32]), we assume that the snake pipefish largely benefitted from the warmer water conditions during the interglacial period, as seen in the present range expansion.

## MATERIAL AND METHODS

### Sampling, DNA extraction, and sequencing

A single individual of *Entelurus aequoreus* (Linnaeus 1758) (NCBI: txid42861, marinespecies.org:taxname: 127379) was caught by trawling during an annual monitoring expedition to the Dogger Bank in the North Sea in July 2021 (trawl start coordinates 54.993633, 2.940833; end coordinates 55.0077, 2.929867) with the permission of the Maritime Policy Unit of the UK Foreign and Commonwealth Office. The study complied with the 'Nagoya Protocol on Access to Genetic Resources and the Fair and Equitable Sharing of Benefits Arising from Their Utilization'. The sample was initially frozen at −20 °C and later stored at −80 °C.

High-molecular-weight genomic DNA was extracted from muscle tissue, following the protocol by Mayjonade *et al.* [33], with the addition of Proteinase K. We evaluated the quantity and quality of the DNA with the Genomic DNA ScreenTape on the Agilent 2200 TapeStation system (Agilent Technologies), as well as with the Qubit® dsDNA BR Assay Kit.

For long-read sequencing, a PacBio SMRT Bell CLR library was prepared using the SMRTbell Express Prep kit v3.0 kit (Pacific Biosciences – PacBio, Menlo Park, CA, USA) and sequenced on the PacBio Sequel IIe platform. A proximity-ligation library was compiled

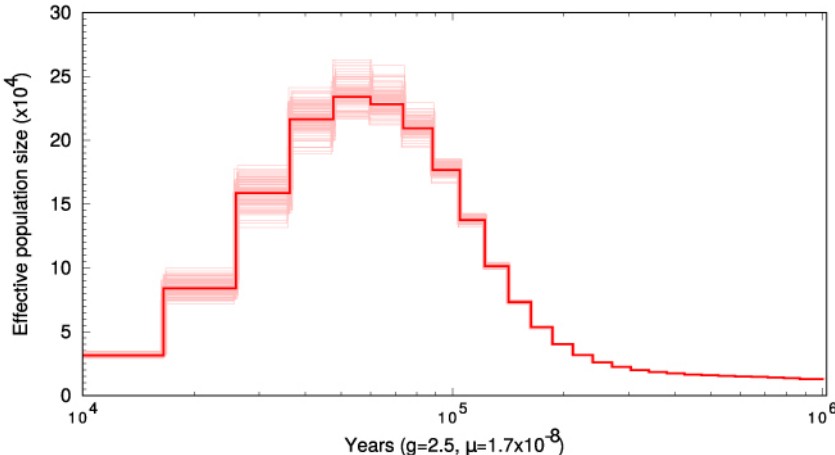

**Figure 3.** Demographic history of the snake pipefish estimated using the PSMC framework. Using a generation time of 2.5 years [72] and a substitution rate of $1.7 \times 10^{-8}$ per site per generation [71], a model was created covering the last 10 kya to 1 Mya. The *x*-axis represents time in number of years ago and the *y*-axis shows the effective population $N_e$ size in tens of thousands of individuals. The model indicates a peak in $N_e$ of 250 thousand individuals during the Pleistocene around 100 thousand years ago.

with muscle tissue following the Dovetail™ Omni-C protocol (Dovetail Genomics, Santa Cruz, California, USA). In addition, a standard whole-genome 150 bp paired-end Illumina library was prepared using the NEBNext Ultra II library preparation kit (New England Biolabs Inc., Ipswich, USA). Finally, the proximity ligation and the paired-end library were shipped to Novogene (UK) for sequencing on the Illumina NovaSeq 6000 platform (RRID:SCR_016387).

### Pre-processing and genome size estimation

The PacBio subreads were converted from BAM into FASTQ format using the PacBio Secondary Analysis Tool BAM2fastx v.1.3.0 [34]. Quality control, trimming, and filtering of the Illumina reads were performed using fastp v0.23.1 (RRID:SCR_016962) [35] with the settings "*-g -3 -l 40 -y -Y 30 -q 15 -u 40 -c -p -j -h -R -w N*". To estimate the genome size of the snake pipefish, we performed *k*-mer profiling using the standard short-read Illumina data. We first ran Jellyfish v2.3.0 (RRID:SCR_005491) [36] to generate a histogram of *k*-mers with a length of 21 bp. Subsequently, we used this data to obtain a genome profile using GenomeScope v2.0 (RRID:SCR_017014) [37]. We further tested alternative *k*-mer lengths between 17- and 25-mers. No significant differences in the estimated genome size were detected except for the 17-mer, which resulted in a smaller genome size estimation of ~500 Mbp.

### Genome assembly and polishing

We assembled the genome from the PacBio long-read data using WTDBG v.2.5 (RRID:SCR_017225) [38]. The resulting assembly was first polished using the PacBio data with Flye v.2.9 (RRID:SCR_017016) [39], using Minimap v.2.17 [40] for mapping. Afterwards, we conducted two rounds of short-read polishing by mapping reads onto the assembly with BWA-MEM v.0.7.17 (RRID:SCR_010910) [41], followed by error correction with Pilon v1.23 (RRID:SCR_014731) [42].

## Assembly quality control and scaffolding

The polished assembly contigs were anchored into chromosome-scale scaffolds utilizing the generated proximity-ligation Omni-C data. First, the data were mapped and filtered to the assembly following the Arima Hi-C mapping pipeline used by the Vertebrate Genome Project [43]. In brief, reads were mapped using BWA-MEM v.0.7.17 [41], the mapped reads were filtered with samtools v.1.14 (RRID:SCR_002105) [44], and the duplicated reads were removed with "MarkDuplicates" in Picard v.2.26.10 (RRID:SCR_006525) [45]. The filtered mapped reads were then used for proximity-ligation scaffolding in YaHs v.1.1 [46]. Gaps in the scaffolded assembly were closed with TGS-GapCloser v.1.1.1 (RRID:SCR_017633) [47] using a subset (25%) of the PacBio subreads due to computational constraints. To further improve the assembly's contiguity, scaffolding and gap-closing were performed a second time using a different subset of PacBio reads for gap-closing. The PacBio read subsets were generated with Seqtk v.1.3 (RRID:SCR_018927) [48] using the random number generator seeds 11 and 18. Gene set completeness was analyzed with BUSCO v.5.4.7 [49] using the Actinopterygii set of core genes (*actinopterygii_odb10*). Assembly continuity was evaluated using QUAST v5.0.2 (RRID:SCR_001228) [50], and mapping rates were assessed by QualiMap v2.2.1 (RRID:SCR_001209) [51]. Finally, BlobToolsKit v.4.0.6 [52] performed contamination screening.

## Repeat landscape analysis and genome annotation

The TE annotation was done in three steps. First, we used RepeatMasker v4.1.5 [53] to annotate and hard-mask known Actinopterygii repeats from Repbase (RRID:SCR_021169), which comprises a database of eukaryotic repetitive DNA element sequences [54]. Secondly, a *de novo* library of TE was created from the hard-masked genome assembly using RepeatModeler v2.0.4 [55], which includes RECON v1.08 (RRID:SCR_021170) [56], RepeatScout v1.0.6 (RRID:SCR_014653) [57], as well as LTRharvest and LTR_retriever (RRID:SCR_018970 and RRID:SCR_017623, respectively) [58, 59]. Finally, predicted repeats were annotated with a second run of RepeatMasker on the hard-masked assembly obtained in the first run. The results of both RepeatMasker runs were then combined. A summary of TEs and the relative abundance of repeat classes in the genome are shown in Table 2 and Figure 2.

The genome was annotated using the BRAKER3 pipeline (RRID:SCR_018964) [60–65], combining a *de novo* gene calling and a homology-based gene annotation. For protein references, we combined the vertebrate-specific protein collection from OrthoDB (RRID:SCR_011980) and the protein collection of the greater pipefish (*Syngnathus acus*) genome [30] made by the NCBI (see: GCF_901709675.1, last accessed 12th October 2023). To further filter genes based on the support of introns and using extrinsic homology evidence, we used TSEBRA [66] with an "intron_support=0.1". The resulting set of proteins was tested for completeness using BUSCO v.5.4.7 [49] in "protein mode" and run against the Actinopterygii-specific set of core genes. Finally, functional annotation was done using InterProScan v5 (RRID:SCR_005829) [67].

## Variant calling and demographic inference

The preprocessed short reads were mapped to the final assembly using BWA-MEM v.0.7.17 [41], followed by the removal of duplicate reads with "MarkDuplicates" in Picard v.2.26.10 [45] and the evaluation of the mapping quality using Qualimap v2.2.1 [51]. Indels



**Table 2.** Repeat content of the genome assembly. Class: class of the repetitive regions. Count: number of occurrences of the repetitive region. bpMasked: number of base pairs masked; %Masked: percentage of base pairs masked. LINE: Long Interspersed Nuclear Elements (include retroposons); LTR: Long Terminal Repeat elements (including retroposons); SINE: Short Interspersed Nuclear Elements; RC: Rolling Circle.

| Class | Count | bpMasked | %masked |
|---|---|---|---|
| ARTEFACT | 4 | 84 | 0.00% |
| DNA | 2,765,297 | 372,407,739 | 22.40% |
| LINE | 850,222 | 167,337,419 | 10.06% |
| LTR | 177,214 | 55,439,687 | 3.33% |
| PLE | 1 | 0 | 0.00% |
| RC | 32,348 | 3,385,084 | 0.20% |
| SINE | 435,464 | 32,709,572 | 1.95% |
| Unknown | 3,628,328 | 534,216,084 | 32.14% |
| Low complexity | 127,733 | 3,095,322 | 0.19% |
| Satellite | 21,221 | 7,145,469 | 0.43% |
| Simple repeat | 1,437,090 | 61,077,339 | 3.67% |
| rRNA | 4,394 | 534,599 | 0.03% |
| scRNA | 5 | 504 | 0.00% |
| snRNA | 695 | 46,845 | 0.00% |
| tRNA | 6,029 | 533,812 | 0.03% |
| Total | 9,486,045 | 1,237,929,559 | 74.93% |

in the BAM files were first identified and then realigned with "RealignerTargetCreator" and "IndelRealigner" as part of the Genome Analysis Toolkit v3.8-1 [68]. Subsequently, samtools v.1.14 [44] was used to check and remove unmapped, secondary, QC-failed, duplicated, and supplementary reads, keeping only reads mapped in proper pairs in non-repetitive regions of the 28 chromosome-scale scaffolds.

Sambamba v 1.0.0 (RRID:SCR_024328) [69] was used to estimate site depth statistics. Minimum and maximum thresholds for the global site depth were set to d ± (5 × MAD), where d is the global site depth distribution median and MAD is the median absolute deviation. Variant calling was performed using the bcftools v1.17 (RRID:SCR_005227) [70] commands "mpileup" and "call" [-m]. Variants were then filtered with bcftools "filter" [-e "DP < d − (5 × MAD) ‖ DP > d + (5 × MAD) ‖ QUAL < 30"], thus removing sites with low quality and out of range depth. Finally, bcftools was used to estimate the genome-wide heterozygosity as the proportion of heterozygous sites using the "stats" command.

Long-term changes in the effective population size ($N_e$) over time were estimated using the PSMC model [17]. This analysis used the diploid consensus genome sequences generated by bcftools v1.17 [70] with the script *"vcfutils.pl"* from the processed BAM files, as described above. Sites with read-depth up to a third of the average depth or above twice each sample's median depth and with a consensus base quality < 30 were removed. PSMC was executed using 25 iterations, employing a maximum $2N_0$-scaled coalescent time of 15, an initial θ/ρ ratio of 5, and 64 atomic time intervals (4 + 25 × 2 + 4 + 6) to infer the scaled mutation rate, the recombination rate, and the free population size parameters, respectively. We performed 100 bootstrap replicates by randomly sampling with replacement 1 Mb blocks from the consensus sequence for all individuals. A mutation rate $\mu$ of $1.7 \times 10^{-9}$ per site per generation [71] and a generation length of 2.5 years [72] were employed for plotting.

## DATA AVAILABILITY

The *de novo* genome and all underlying raw data were uploaded to NCBI under the BioProject PRJNA1005573, BioSample SAMN36988691, genome assembly GCA_034508595.

All other data, including the repeat and gene annotation, is available in the GigaDB repository [73].

## ABBREVIATIONS

CLR, continuous long reads; kya, thousand years ago; MAD, median absolute deviation; PSMC, Pairwise Sequentially Markovian Coalescent; TE, transposable element.

## DECLARATIONS

### Ethics approval and consent for publication

Not Applicable.

### Competing interests

The authors declare that they have no competing interests.

### Author contributions

MW, BF, MS, AJ, and SW designed the study. SW, JR, HMI, JM, CDS, LG, MJW, HO, and MWI performed laboratory procedures and sequencing. MW, BF, RC, MDJ, MN, DP, YS, KZ, JR, HMI, JM, CDS, LG, MJW, HO, MWI, MAN, and SW conducted bioinformatic processing and analyses. All authors contributed to writing this manuscript.

### Acknowledgements

The present study is a result of the Centre for Translational Biodiversity Genomics (LOEWE-TBG) and was supported through the program 'LOEWE-Landes-Offensive zur Entwicklung Wissenschaftlich-ökonomischer Exzellenz' of Hesse's Ministry of Higher Education, Research, and the Arts.

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
