## [Editor Report]

Editor’s AssessmentThe snake pipefish, Entelurus aequoreus, is a species of fish that dwells in open seagrass habitats in the northern Atlantic. As a pipefish, it is a member of the Syngnathidae family of fish which also includes seahorses and seadragons. In recent years it has expanded its population size and range into arctic waters. To better understand these demographic changes genomic data is useful, and to address this a high-quality reference genome has been produced. Building on a previous short-read reference, a near chromosome-scale genome assembly for the snake pipefish was assembled using PacBio CLR and Hi-C reads. After revisions the authors provided more details on the assembly metrics, the final assembly has a length of 1.6 Gbp, with scaffold and contig N50s of 62.3 Mbp and 45.0 Mbp respectively. Demographic inference analysis of the snake pipefish genome using this data enables tracing of population changes over the past 1 million years, and this reference will allow further analyses and studies relating these to changes in climate.

---

## [Reviewer Report]

Reviewer name and names of any other individual's who aided in reviewer Sarah FlanaganDo you understand and agree to our policy of having open and named reviews, and having your review included with the published papers. (If no, please inform the editor that you cannot review this manuscript.)YesIs the language of sufficient quality?YesPlease add additional comments on language quality to clarify if needed
Are all data available and do they match the descriptions in the paper? NoAdditional CommentsI received an NCBI link which took me to the raw data files and a BioSample description, but it did not link to the assembled and annotated genome. Are the data and metadata consistent with relevant minimum information or reporting standards? See GigaDB checklists for examples <a href="http://gigadb.org/site/guide" target="_blank">http://gigadb.org/site/guide</a>YesAdditional CommentsIs the data acquisition clear, complete and methodologically sound?YesAdditional CommentsIs there sufficient detail in the methods and data-processing steps to allow reproduction?YesAdditional CommentsOnly one point was not clear to me in the methods -- please clarify in the text which data was used to generate consensus genome sequences using vcfutils (lines 240-241). How did this differ from the assembled and annotated genome?  
Is there sufficient data validation and statistical analyses of data quality? YesAdditional CommentsIs the validation suitable for this type of data?YesAdditional CommentsIs there sufficient information for others to reuse this dataset or integrate it with other data?YesAdditional CommentsAny Additional Overall Comments to the AuthorIn the abstract and introduction, the description of the habitat of the species is confusing and it was not clear from the manuscript as written that there are two ecotypes, one that is pelagic and one that is coastal. Consider re-phrasing these sections (lines 31-32, 57-59, and 61-62) to better describe the habitat of this species.   Please also consider increasing the font size of the labels in Figure 1 -- the details are very difficult to read. RecommendationMinor Revision

---

## [Reviewer Report]

Reviewer name and names of any other individual's who aided in reviewer Yanhong ZhangDo you understand and agree to our policy of having open and named reviews, and having your review included with the published papers. (If no, please inform the editor that you cannot review this manuscript.)YesIs the language of sufficient quality?YesPlease add additional comments on language quality to clarify if needed
Are all data available and do they match the descriptions in the paper? NoAdditional CommentsThere is no BioProject available for review at the link.Are the data and metadata consistent with relevant minimum information or reporting standards? See GigaDB checklists for examples <a href="http://gigadb.org/site/guide" target="_blank">http://gigadb.org/site/guide</a>NoAdditional Comments"the GigaDB repository:DOI:XXXXX." I am not sure that the authors have upload the data.
Is the data acquisition clear, complete and methodologically sound?NoAdditional CommentsI am not sure that the authors have upload the data.Is there sufficient detail in the methods and data-processing steps to allow reproduction?YesAdditional CommentsIs there sufficient data validation and statistical analyses of data quality? NoAdditional CommentsI need more information.Is the validation suitable for this type of data?NoAdditional CommentsI need more information.Is there sufficient information for others to reuse this dataset or integrate it with other data?NoAdditional CommentsI need more information.Any Additional Overall Comments to the AuthorIn line 41, you mean “50-100 kya”?  The authors need to provide more details about the genomic data： Genome size estimation based on K-mer spectrum？ Statistics of genomic characteristics from K-mer? Statistics of Hi-C sequencing raw data, such as raw bases, clean bases. Statistics of the pseduchromosome assemblies using Hi-C data. The result of BUSCO assessment, how about complete BUSCOs? complete single-copy？ Statistics of gene predictions in the snake pipefish Statistics of the noncoding RNA in the snake pipefish genome.  The author claims that all other data, including the repeat and gene annotation, was uploaded to the GigaDB repository: DOI: XXXXX. I can not find these data. “DOI: XXXXX”? What does that mean?
RecommendationMajor Revision